# In Silico and In Vitro Study towards the Rational Design of 4,4′-Disarylbisthiazoles as a Selective α-Synucleinopathy Biomarker

**DOI:** 10.3390/ijms242216445

**Published:** 2023-11-17

**Authors:** Bright C. Uzuegbunam, Junhao Li, Wojciech Paslawski, Wolfgang Weber, Per Svenningsson, Hans Ågren, Behrooz Hooshyar Yousefi

**Affiliations:** 1Department of Nuclear Medicine, Technical University of Munich, 81675 Munich, Germany; 2Department of Physics and Astronomy, Uppsala University, 751 20 Uppsala, Sweden; 3Department of Clinical Neuroscience, Karolinska Institute, 171 76 Stockholm, Sweden; 4Department of Nuclear Medicine, Philipps University of Marburg, 35043 Marburg, Germany

**Keywords:** α-synuclein aggregates, 4,4′-diarylbisthiazole, binding affinity, Hantzsch thiazole synthesis, lipophilicity, O-fluoroethylation, O-fluoroPEGylation, O-methylation

## Abstract

The α-synucleinopathies are a group of neurodegenerative diseases characterized by the deposition of α-synuclein aggregates (α-syn) in the brain. Currently, there is no suitable tracer to enable a definitive early diagnosis of these diseases. We reported candidates based on 4,4′-disarylbisthiazole (DABTA) scaffold with a high affinity towards α-syn and excellent selectivity over Aβ and tau fibrils. Based on prior in silico studies, a focused library of 23 halogen-containing and O-methylated DABTAs was prepared. The DABTAs were synthesized via a modified two-step Hantzsch thiazole synthesis, characterized, and used in competitive binding assays against [^3^H]PiB and [^3^H]DCVJ. The DABTAs were obtained with an overall chemical yield of 15–71%, and showed a calculated lipophilicity of 2.5–5.7. The ligands demonstrated an excellent affinity to α-syn with both [^3^H]PiB and [^3^H]DCVJ: K_i_ 0.1–4.9 nM and up to 20–3900-fold selectivity over Aβ and tau fibrils. It could be concluded that in silico simulation is useful for the rational design of a new generation of DABTAs. Further investigation of the leads in the next step is encouraged: radiolabeling of the ligands with radioisotopes such as fluorine-18 or carbon-11 for in vivo, ex vivo, and translational research and for further in vitro experiments on human-derived protein aggregates.

## 1. Introduction

The deposition of α-synuclein aggregates (α-syn) in different brain regions is a neuropathological hallmark that defines a group of a heterogeneous group of neurodegenerative diseases (NDD) known as the α-synucleinopathies. These include the Lewy body diseases (LBD) such as Parkinson’s disease (PD) and dementia with Lewy bodies (DLB), which are characterized by the intraneuronal deposition of α-syn in neuropathological lesions termed Lewy bodies (LB) and Lewy neurites (LN). In multiple system atrophy (MSA), a non-LBD α-synucleinopathy, the α-syn is deposited in the oligodendrocytes as glial cytoplasmic inclusions [1,2,3]. 

Clinical presentations of these α-synucleinopathies are believed to be preceded by the aggregation of α-synuclein and its occultic deposition which drives neuronal dysfunction followed by death. For instance, in PD, motor symptoms become apparent after the loss of nearly 60–80% of the dopaminergic neurons (DN) of the SNpc [4], as well as nerve cell loss in the nucleus basalis of Meynert (NBM), locus coereleus (LC), and the dorsal motor nucleus of the vagus (DMV) [5]. Thus, the early and differential diagnosis of these groups of NDD advocates the development of in vivo imaging agents, which would not only enable an early diagnosis, but also the tracking of disease progression as well as the monitoring of the efficacy of therapy. 

With the aid of positron emission tomography (PET) and single-photon emission computed tomography (SPECT), imaging agents with adequate affinity and pharmacokinetic properties could enable in vivo imaging of the α-syn, when labeled with either positron or single-photon-emitting radioisotopes for the respective modalities [3]. 

The development of α-syn tracers is faced with general pitfalls that characterize the development of brain tracers such as molar weight (MW). The brain uptake of ligands correlates inversely to the square root of its MW [6]. Another factor is lipophilicity, which plays an increased role for the α-syn tracers, since the location of α-syn is majorly intraneuronal or intraglial. Therefore, the tracers ought to be sufficiently lipophilic to partition both across the blood–brain barrier (BBB) as well as the neuron cell membrane [7,8]. However, the tracers should not be too lipophilic that brain clearance of the nonspecifically bound brain tracers is compromised, which complicates the kinetic modeling, quantification and the reproducibility [2,9]. 

Furthermore, the development of α-syn radiotracers is replete with its own unique problems. Unfortunately, α-syn usually co-localizes with other protein aggregates that are also present in other NDD such as Aβ [7] and tau [10]. Hence, selectivity is a major challenge: β-sheet stacking is a common pathway to the aggregation of proteins, which results in similar β-pleated sheets common to the above-mentioned protein aggregates. Therefore, the prospective tracer must bind to α-syn in the nanomolar range, if possible, even in the subnanomolar range. The development of α-syn radiotracers is further complicated by the lower absolute concentration of α-syn compared to Aβ and tau [2,7,8,11]. 

Despite the above-mentioned limitations, several research groups are working on developing an ideal α-syn tracer with increasing success [12,13]. Our contribution is based on a DABTA scaffold (Figure 1a). Preliminary in vitro binding assays indicated that the presence of the methylenedioxy functional group (Figure 1b) is essential to increased affinity of the DABTAs to α-syn and selectivity over Aβ and tau. For this reason, all the ligands reported in this paper bear this functional group as well. 

In an earlier collaboration with the coauthors of this paper, some DABTA derivatives with varied affinity to α-syn and selectivity over Aβ and tau were designed. To accomplish this, several in silico modeling techniques were applied, such as molecular docking, MM/GBSA, free-energy calculations and metadynamics stimulation through which the interactions between the DABTAs and the fibrils (α-syn, Aβ, and tau) were studied [1]. As a result, the advantages of some chemical moieties and functional groups were further determined. 

This present study focused on the lipophilicity and binding affinity of the DABTAs to α-syn, Aβ and tau as a function of their chemical moieties and functional groups. 

## 2. Results

### 2.1. Chemical Yields (CYs)

The CYs were moderate to high and reflected the success of the step I in the chemical synthesis, especially with regard to semipreparative HPLC purification. The repeated synthesis of the 4-aryl/heteroarylthiazole-2-carbothioamide intermediates (b) enabled the enactment of better purification strategies to obtain them in higher CYs. The CYs of (b) obtained in step I of the chemical synthesis are provided in the Materials and Methods (Section 4.1). In Table 1 are presented: the CYs of Step II with respect to Step I, Step III (for some compounds) and the overall CYs. The structures of all the ligands used for the competitive binding assays and in silico studies are presented in Figure 2.

### 2.2. Competitive Binding Assays and clogP 

All the reported DABTAs were sufficiently lipophilic with clogP between 2.5 and 5.7. As expected, the DABTAs bearing the benzodioxole ring were more lipophilic than their [1,3]dioxolo[4,5-b]pyridine ring-containing analogues (Figure 2): **d_4_**/**d_2_**, **d_8_**/**d_6_**, **f_4_**/**f_5_**, **f_8_**/**f_6_**, **f_13_**/**f_7_**, and **f_15_**/**f_16_**.

Although a high affinity to α-syn was observed, the ligands showed varied affinity to the protein aggregates depending on the ligand against which they were competing, which in this case were [^3^H]DCVJ and [^3^H]PiB. Using [^3^H]DCVJ, an interesting tendency was observed, namely, that DABTAs that bear the [1,3]dioxolo[4,5-b]pyridine moiety showed higher affinity to α-syn than their benzodioxole-ring-bearing counterparts (Table 2, Appendix A), except in the case of **f_6_**. Although **f_6_** bears no [1,3]dioxolo[4,5-b]pyridine ring, it still showed subnanomolar affinity to α-syn, a property that could be attributed to the fluoroethylethoxy functional group it bears. Although this tendency was also observed using [^3^H]PiB, it was inconsistent with slight reduction in affinity and selectivity (Table 2, Appendix A).

Due to the low solubility of **f_8_**, **f_16_**, **f_18_**, and **h_13_** in the incubation medium, it was difficult to perform the binding assays with them: it was impossible with **f_16_** at any concentration; for **f_8_**, **f_18_**, and **h_13_**, it was only possible at the low ligand concentration at which the α-syn binding assays were carried out (Table 2).

### 2.3. In Silico Modeling of the Binding of DABTAs to α-syn

We docked ligands **f_4_** to **f_20_** and **h_13_** to the binding sites (sites 1–4, Figure 3A) identified from a previous study [14], followed by prime MM/GBSA binding free-energy calculations. The sites with the lowest binding free energies for each DABTA are reported in Figure 3B, which shows that DABTAs depending on the functional groups borne show varied preferences to the identified sites.

The three surface sites which are sites-1, -2, and -4 bear at least one basic amino acid residue. Interestingly, the DABTAs exhibit a very competitive preference for site-2 (7 compounds, **f_4_**, **f_7_**, **f_9_**, **f_10_**, **f_14_**, **f_18_,** and **f_19_**) of all the surface sites compared to the inner cavity site-3 (8 compounds, see Figure 3B), which is more similar to a conventional ligand binding pocket. Interestingly, the two DABTAs that simultaneously bear two fluorine atoms **f_10_** and **f_20_** showed different binding properties. On one hand, **f_10_**, which bears both a fluorine atom directly bonded to the aromatic ring and another linked to its core DABTA scaffold via a PEG linker **(**Figure 2), bound to site-2 with the highest binding free energy (−76.8 kcal/mol) observed in the DABTAs. On the other hand, **f_20_**, whose both fluorine atoms are linked to its core DABTA scaffold via a PEG linker **(**Figure 2) bound to site-1, actually the only DABTA that bound appreciably to this site, with 21.8 kcal/mol less energy. Although this difference in binding affinity might be due to the different spatial characteristics of the ligands, the strong negative inductive and positive mesomeric effects of the highly electronegative ring-bound Csp2-fluorine atom in **f_10_** might have also played a role. This difference in the binding properties of both ligands as seen in the in silico study was corroborated by the binding assays (Table 2), in which the K_i_ of **f_10_** to α-syn was higher and roughly the same with both [^3^H]DCVJ and [^3^H]PiB.

Based on analysis of the binding modes of the DABTAs as revealed by the docking experiments, it was seen that all the DABTAs adopted stretched modes with their axes parallel to that of the protofibril upon binding (Figure 4). Such a docking interaction mode has also been observed in other in silico studies carried out with ligands that bind to protofibrils from neurodegenerative diseases.

Further, 100-ns simulations were performed for each ligand bound to each docking-derived site (surface sites-1, -2, and -4 and the interior cavity site 3). From the root-mean-square deviation (RMSD) values, the mobility of ligands on the surface site was observed to be significantly larger than those in the cavity site (Appendix A). 

To determine the kind of interaction preferred by the DABTAs in their interactions with the surface sites of α-syn, a simulation interaction analysis was carried out in which it was discovered that, although all the DABTAs bear four aromatic rings, π–π interaction is not the main interaction in ten of the eighteen evaluated ligands (Figure 5A). The overall accumulated percentages of interactions follow the order–hydrophobic interaction, π–π interaction, cation–π interaction, and hydrogen bonding from the trajectories of most ligands starting from the three surface sites. 

In the MD simulations, where the interactions between the DABTAs and α-syn were narrowed down to the residues in the surface sites, it was clearly seen π–π interactions played an important role in the interaction of most of the DABTAs with the histidine (H50) and phenylalanine (F94) residues (Figure 5B), which is hardly surprising since they all bear aromatic and heteroaromatic rings. Hydrophobic interactions with the amino acid residues of hydrophobic amino acids such as I88, V48, V52, F94, and A85 were observed in the DABTAs (**f_4_**, **f_5_**, **f_8,_ f_10,_ f_11_**, **f_14_**, **f_15_**, **f_16_**, **f_17_**, and **f_20_**), with overall accumulated percentage over 70% (Figure 4)**_._** Unexpectedly, **f_10_** surpassed all the DABTAs with respect to hydrogen bond interactions with lysine residues (K96) in the surface sites (sites-1, -2, and -4) of α-syn (Figure 5A,B), while the DABTAs with a relatively high tPSA, **f_13_** and **h_13_**, performed poorly in this aspect. 

Moreover, it was also interesting to observe that the majority of the pyridine-bearing DABTAs, which are expected to form stronger hydrogen bonds or did not live up to the expectation, they counterintuitively partook more in hydrophobic interactions in the simulations. Cation–π interaction was seen not only in the DABTAs bearing electron-rich aromatic rings with protonated lysine residues (K45) but also in those with only heteroaromatic rings such as **f_4_**, **f_13_**, and **f_16_**. The combination of high π–π/cation–π interactions observed for these only-heteroaromatic-rings-bearing DABTAs might have reduced their tendency to form hydrogen bonds since π–π/cation–π interactions require a specific angle between aromatic/heteroaromatic rings of the DABTAs and the corresponding amino acid residues. During the formation of these angle-specific interactions, it is possible that initially formed hydrogen bonds were broken.

## 3. Discussion

The strategies employed in the CY of the DABTAs were discussed in detail in a previous publication [1]. Nevertheless, care was taken to obtain the intermediates (Figure 1), that is, the 4-aryl/heteroarylthiazole-2-carbothioamides in high HPLC purity. This facilitated an easier purification of the more difficult-to-dissolve downstream products (**d**) and (**f**) (Figure 2 and Figure 3).

The lipophilic profile of the ligands is reminiscent of that of some of the already reported ligands such as **d_6_** and **d_8_**, which showed excellent brain uptake and washout [1]. Even the ligands with relatively higher clogP made up for this by their relatively high tPSA, which might lower unspecific binding both in the periphery and in the brain. 

Although PEGylation contributed to the reduction of lipophilicity (**d_4_**/**f_9_**) (Table 1), a noticeable reduction was observed by replacing the benzene ring in the benzodioxole moiety with a pyridine ring, as was seen in the cases of **d_2_**/**d_4_**, **d_6_**/**d_8_**, **f_5_**/**f_4_**, **f_8_**/**f_6_**, **f_7_**/**f_13_**, and **f_15_**/**f_16_**. A more noticeable reduction of lipophilicity was observed when the benzene ring in the non-benzodioxole moiety (Figure 1b) in the DABTAs was exchanged with a pyrimidine ring, as was seen in **f_5_**/**f_11_** and **f_12_**/**f_13_**. Moreover, the pyrimidine ring in the fluoroethyl/PEGylated DABTAs is positioned in such a way that hydrogen bond formation in the in vivo aqueous medium is minimized, thus facilitating brain permeability of the tracers. Despite the overall reduced lipophilicity, hydrophobic and π–π interactions still dominated the interactions between ligands and the solvent-exposed surface sites (Figure 4). 

The introduction of polar functional groups theoretically reduced the lipophilicity of the DABTAs (Table 2). In practice, however, in some cases, there was decreased solubility, especially for the DABTAs bearing a pyridine and a pyrimidine ring (**d_10_**, **f_8_**, **f_16_**, and **h_13_**) simultaneously, except for **f_18_** which bears only the former. Solubility was so low that some of them had to be excluded from some of the binding assays. The reason for this observation cannot be explained based on the present study, but it is speculated that increased intramolecular π–π stacking in the presence of heteroaromatic rings might have contributed to this phenomenon [15,16,17,18]. 

As already mentioned [1], in preliminary binding assays it was established that the methylenedioxy group is essential for a high affinity to α-syn and selectivity over Aβ and tau. For this reason, this functional group is present in all the DABTAs. Furthermore, the ligands, which are soluble enough to participate in the competitive binding assays, showed a functional-group-dependent affinity to α-syn in the assays, especially with [^3^H]DCVJ. The addition of further heteroaromatic rings to the DABTA scaffold (Figure 1b) might have improved affinity to α-syn due to increased intermolecular π–π stacking with the aromatic rings of aromatic amino acid residues in the binding sites [15,18,19]. For instance, introduction of the pyridine ring in the DABTAs appears to improve affinity to α-syn compared to the ligands without it: good examples are **d_4_** and **d_2_**; **d_8_** and **d_6_**; **f_4_** and **f_5_**; **f_13_** and **f_7_**; **f_17_** and **f_15_**. The pyrimidine takes on this role in the absence of the pyridine ring, as was seen in **f_5_** and **f_11_**. 

This tendency was slightly diminished using [^3^H]PiB, which might be partly explained by the contents of the incubation buffer used in this case, with which the ligands might have interacted better. Moreover, the difference in binding sites on the protein aggregates might be another factor, which is reflected in the binding free-energy calculations for different sites. The difficulty in matching the calculated free energies with the experimental K_i_ values can be related to different experimental and theoretical conditions. The low solubility of some of the DABTAs, as already mentioned, might have played a role in their ability to interact with the protein aggregates. The most interesting of these poorly soluble DABTAs is **f_16_**, which, in spite of its hydrogen-bond-forming functional groups, could not be dissolved in the incubation buffer for the binding experiments. 

Computer simulation is a powerful tool which compensates for the insufficient experimental knowledge of the dynamic and atomic details of the binding of PET tracers to pathologically aggregated protein filaments. Although the binding of small molecules in the cavity site of lipidic α-syn can be observed by NMR spectroscopy [20], the surface site in the fibril may still be more accessible to the ligands in vivo, since the fibrils may twist themselves around each other, thus blocking the entrance to the cavity sites. 

In this study, the surface site-2, which contains more than two aggregated residues, was found to be more preferred by the DABTAs compared to the cavity site-3 (Figure 3B). Notably this was not observed in a recent study of the interaction of some tau PET tracers with the surface sites of 4R tau protofibrils [21]. The mobility of the DABTAs on the surface sites of α-syn are also less significant than other PET tracers on tau protofibrils [21]. 

PEGylation can either lengthen the molecule, when it is added to the para-position of the benzene/pyrimidine ring, or partially bifurcate the molecule when it is located on the meta- or ortho-position. From the interaction analysis, it was seen that the 2–3-repeat PEGylation at the para-position is beneficial for π–π interactions with H50 at site-4 or F94 at site-2 (**f_6_**, **f_13_**, and **h_13_** in Figure 5). For instance, the 3-repeat PEGylation at the para-position in **f_13_** seems to greatly enhance its affinity to α-syn (K_i_ 0.38 nM).

For the branched PEGylation with 3 repeats, the portion of hydrophobic interaction increased significantly (**f_14_** and **f_20_** in Figure 5). The branched PEG moiety could hook some “pop-out” backbones area with the direction perpendicular to the axis of the protofibril. For example, the PEG moiety at the meta-position was found to hook the “pop-out” area, which comprised G67, G68, and A69 (Figure 5), and thereby increase the hydrophobic interactions during the MD simulations. However, the 2-repeat PEGylation at the meta-position increases hydrogen bonding with K45 for ligand **f_9_** (Figure 5). Contact analysis based solely on the distances between any of the non-hydrogen atoms of the ligand and the protein for the surface sites 1, 2, and 4 showed that the best contact residue is N65 for all the ligands except **f_6_**, **f_7_**, **f_13_**, **f_1_**_5_, **f_19_**, and **f_20_**, which mostly contacted with H50 (Appendix A). Despite the fact that the ligands in site-2 have comparable free energies of binding to site-3 (the cavity site generally has a higher affinity for the ligand), a greater extent of contact was made with N65 in the boundary of site-1 during the MD simulations (Figure 3A), which correlated with smaller RMSD values for most ligands in site-1 than in site-2 (Figure 4).

## 4. Materials and Methods

### 4.1. Chemical Synthesis

The materials and instrumentation used, unless otherwise stated, have been previously described [1]. The synthesis procedures of the DABTAs and the reaction schemes are presented below, with the ^1^H and ^13^C NMR spectra of the compounds provided in the Appendix A section.

#### 4.1.1. General Procedure for the Synthesis of the 4-Aryl/heteroarylthiazole-2-carbothioamide Intermediate, **b**


To 1.5 mL of dimethylformamide (DMF) solution of dithiooxamide (DTO) 180.3 mg (1.5 mmol equiv) in a 10 mL vial while stirring was added dropwise the corresponding phenacyl/heteroacylbromide or a 3,4-methylenedioxy(phenacylbromide) (a) (1 mmol) in 500 µL of DMF and left to react overnight at room temperature (rt) (Figure 1). Deviations from the general procedure are specified. 

##### 4.1.1.1. 4-(Benzo[d][1,3]dioxol-5-yl)thiazole-2-carbothioamide, **b_1_**


**b_1_** was synthesized according to the general procedure using 180.3 mg DTO and 243.1 mg of 1-(benzo[d][1,3]dioxol-5-yl)-2-bromoethan-1-one. Semipreparative HPLC purification (SHP) was carried out with 30% aqueous methanol (MeOH) solution, 0.1% tetrahydrofuran (THF), 0.1% trifluoroacetic acid (TFA) (solvent A) and acetonitrile (MeCN), and 0.1% TFA (solvent B) at a flow rate of 5 mL/min and a gradient A/B: 80:20 to 18:82 in 17 min. **b_1_** was obtained in 64.7 ± 3.86%, (n = 5) yield, with ≥99% HPLC purity. [M + 1] 265.0.

##### 4.1.1.2. 4-(3-Hydroxy-4-methoxyphenyl)thiazole-2-carbothioamide, **b_3_**

**b_3_** was synthesized according to the general procedure using 180.3 mg DTO and 245.1 mg of 2-bromo-1-(3-hydroxy-4-methoxyphenyl)ethan-1-one. SHP was carried out with 30% aqueous MeOH solution, 0.1% THF, 0.1% TFA (solvent A) and MeCN, and 0.1% TFA (solvent B) at a flow rate of 5 mL/min and a gradient A/B: 95:5 to 23:77 in 17 min. **b_3_** was obtained in 75.6 ± 4.37%, (n = 2) % yield, with ≥99% HPLC purity. [M + 1] 267.0. 

##### 4.1.1.3. 4-(3-Fluoro-4-methoxyphenyl)thiazole-2-carbothioamide, **b_4_**

**b_4_** was synthesized according to the general procedure using 180.3 mg DTO and 247.1 mg of 2-bromo-1-(3-fluoro-4-methoxyphenyl)ethan-1-one. SHP was carried out with an aqueous mixture of THF and MeOH solution (20%:10%), 0.1% TFA (solvent A) and MeCN, and 0.1% TFA (solvent B) at a flow rate of 5 mL/min and a gradient A/B: 70:30 to 32:68 in 17 min. **b_4_** was obtained in 60.0 ± 1.04%, (n = 2), with ≥99% HPLC purity. [M + 1] 296.1. 

##### 4.1.1.4. 4-(6-Fluoropyridin-3-yl)thiazole-2-carbothioamide, **b_8_**

**b_8_** was synthesized according to the general procedure using 180.3 mg DTO and 219.0 mg of 2-bromo-1-(6-fluoropyridin-3-yl)ethan-1-one. SHP was carried out with an aqueous mixture of THF and methanol solution (0.2%:40%), 0.1% TFA (solvent A) and MeCN, and 0.1% TFA (solvent B) at a flow rate of 5 mL/min and a gradient A/B: 90:10 to 25:75 in 18 min. **b_8_** was obtained in 88.2%, n = 1 yield, with ≥99% HPLC purity. [M + 1] 240.0.

##### 4.1.1.5. 4-(2-Fluoropyrimidin-4-yl)thiazole-2-carbothioamide, **b_10_**

To 1.0 mL of DMF solution of DTO (62.9 mg, 0.5243 mmol) in a 10 mL vial while stirring was added dropwise 2-bromo-1-(2-fluoropyrimidin-5-yl)ethan-1-one (82.3 mg, 0.3495 mmol) in 0.5 mL of DMF. The mixture was left to run overnight at rt. SHP was carried out with 30% aqueous MeCN solution, 0.1% TFA (solvent A) and MeCN, and 0.1% TFA (solvent B) at a flow rate of 5 mL/min and a gradient A/B: 92:08 to 22:78 in 17 min. **b_10_** was obtained in 32%, n = 1 yield, with ≥98% HPLC purity. [M + 1] 241.2.

##### 4.1.1.6. 4-(5-Hydroxypyrimidin-2-yl)thiazole-2-carbothioamide, **b_11_**

**b_11_** was synthesized according to the general procedure using 180.3 mg DTO and 217 mg of 2-bromo-1-(5-hydroxypyrimidin-2-yl)ethan-1-one. SHP was carried out with 30% aqueous MeOH solution, 0.1% THF, 0.1% TFA (solvent A) and MeCN, and 0.1% TFA (solvent B) with a flow rate of 5 mL/min and a gradient A/B: 90:10 to 30:70 in 17 min. **b_11_** was obtained in 73.3 ± 2.42%, (n = 2) yield, with ≥99% HPLC purity. [M + 1] 239.2.

##### 4.1.1.7. 4-(3-Fluoro-4-hydroxyphenyl)thiazole-2-carbothioamide, **b_12_**

To 1.0 mL of DMF solution of DTO (116.0 mg, 0.9655 mmol) in a 10 mL vial while stirring was added dropwise 2-bromo-1-(3-fluoro-4-hydroxyphenol)ethan-1-one (150 mg, 0.6436 mmol) in 1.0 mL of DMF. The mixture was left to run overnight at rt. SHP was carried out with 30% aqueous MeOH solution, 0.1% THF, 0.1% TFA (solvent A) and MeCN, and 0.1% TFA (solvent B) with a flow rate of 5 mL/min and a gradient A/B: 90:10 to 22:78 in 16 min. **b_12_** was obtained in 76.1 ± 2.51%, (n = 3) yield, with ≥99% HPLC purity. [M + 1] 255.2.

##### 4.1.1.8. 4-(4-Hydroxyphenyl)thiazole-2-carbothioamide, **b_13_**

To 0.5 mL of DMF solution of DTO (83.84 mg, 0.6975 mmol) in a 10 mL vial while stirring was added dropwise 2-bromo-1-(4-hydroxyphenyl)ethan-1-one (100 mg, 0.4650 mmol) in 0.5 mL of DMF. The mixture was left to run overnight at rt. SHP was carried out with 30% aqueous MeOH solution, 0.1% TFA, 0.1% THF (solvent A) and MeCN, and 0.1% TFA (solvent B) with a flow rate of 5 mL/min and a gradient A/B: 70:30 to 20:80 in 18 min. **b_13_** was obtained in 72.2%, (n = 1) yield, with ≥99% HPLC purity. [M + 1] 237.1. 

##### 4.1.1.9. 4-(2,4-Dihydroxyphenyl)thiazole-2-carbothioamide, **b_16_**

To 1.0 mL of DMF solution of DTO (120 mg, 1.5 mmol) in a 10 mL vial while stirring was added dropwise 4-(2,4-dihydroxyphenyl)thiazole-2-carbothioamide (100 mg, 0.4328 mmol) in 0.5 mL of DMF. The mixture was left to run overnight at rt. SHP was carried out with 30% aqueous MeOH solution, 0.1% TFA, 0.2% THF (solvent A) and MeCN, and 0.1% TFA (solvent B) with a flow rate of 5 mL/min and a gradient A/B: 95:05 to 16:84 in 17 min. **b_13_** was obtained in 70.6%, (n = 1) yield, with ≥99% HPLC purity. [M + 1] 253.

#### 4.1.2. General Procedure for the Synthesis of the Asymmetric DABTAs, **d**

To 1.0 mL of DMF solution of another phenacyl/heteroacylbromide or a 3,4-methylenedioxy (heteroacylbromide) (1.3 mmol equiv) (Figure 2) in a 10 mL vial while stirring was added the corresponding 4-aryl/heteroarylthiazole-2-carbothioamides intermediates (**b**) in 500 µL of DMF (quantity is specified for each ligand) and stirred overnight. A solid precipitate (the product, **d**) is formed as a result (deviations will be stated). Centrifugation carried out during trituration were performed at 10 °C. All the ligands were dried in vacuo to remove the organic solvents followed by lyophilization in order to remove residual water. The reaction schemes are shown below in Figure 2A,B. 

##### 4.1.2.1. 5-(4′-(Benzo[d][1,3]dioxol-5-yl)-[2,2′-bithiazol]-4-yl)-2-methoxyphenol, **d_1_**

**d_1_** was synthesized according to the general procedure described in (Section 4.1.2). First, 193.1 mg of 2-bromo-1-(3-hydroxy-4-methoxyphenyl)ethan-1-one dissolved in 500 µL of DMF was added to 160.0 mg (0.6060 mmol) of **b_1_** dissolved in 500 µL of DMF in a 10 mL glass vial. The mixture was stirred overnight at 50 °C. The resulting mixture was transferred into a 10 mL falcon tube, 5 mL of methanol (MeOH) was added, and the mixture was centrifuged at 6000 rpm for 10 min. The resulting sediment was then dissolved in DMSO. Both the supernatant and the dissolved sediment in DMSO solution were then purified using the semipreparative HPLC. SHP was carried out with 20% aqueous solution of THF, 0.1% TFA (solvent A) and MeCN, 0.1% TFA (solvent A) with a flow rate of 5 mL/min and a gradient A/B: 40:60 to 14:86 in 18 min. **d_1_** was obtained as a pale yellow amorphous solid after solvent removal with ≥99% purity. [M + 1] 411.1.

##### 4.1.2.2. 4-(Benzo[d][1,3]dioxol-5-yl)-4′-(3-fluoro-4-methoxyphenyl)-2,2′-bithiazole, **d_2_**

**d_2_** was synthesized according to the general procedure (Section 4.1.2). 15.6 mg bromo-1-(3-fluoro-4-methoxyphenyl)ethan-1-one dissolved in 500 µL of DMF was added to 20 mg (0.04849 mmol) of **b_1_** dissolved in 500 µL of DMF solution in a 10 mL vial. The mixture was stirred overnight at 45 °C. The resulting mixture was transferred into a 10 mL falcon tube and centrifuged at 6000 rpm for 10 min. The resulting sediment was sonicated twice in 5 mL of MeOH for 10 min, each time followed by centrifugation for 10 min (6000 rpm) till 99% HPLC purity of **d_2_**. **d_2_** was obtained as a pale violet amorphous solid after solvent removal. 

##### 4.1.2.3. 5-(4′-([1,3]Dioxolo[4,5-b]pyridin-6-yl)-[2,2′-bithiazol]-4-yl)-2-methoxyphenol, **d_3_**

**d_3_** was synthesized according to the general procedure (Section 4.1.2). First, 188.7 mg of 1-([1,3]dioxolo[4,5-b]pyridin-6-yl)-2-bromoethan-1-one dissolved in 500 µL of DMF acidified with 25 μL of glacial acetic acid was added to 158.4 mg (0.5948 mmol) of **b_3_** dissolved in 500 µL of DMF acidified with 25 μL of glacial acetic acid in a 10 mL glass vial. The mixture was stirred overnight at 50 °C. The resulting mixture was transferred into a 10 mL falcon tube, and the mixture was centrifuged at 6000 rpm for 10 min. The resulting sediment was sonicated twice in 5 mL of MeOH for 10 min, each time followed by centrifugation for 10 min, yielding **d_3_** with 99% HPLC purity. The resulting supernatants were combined and purified using semiprep HPLC. SHP was carried out with 30% aqueous MeOH solution, 0.4% THF, 0.1% TFA (solvent A) and MeCN, and 0.1% TFA (solvent A) with a flow rate of 5 mL/min and a gradient A/B: 80:20 to 14:86 in 19 min. Overall, **d_3_** was obtained with ≥99% HPLC purity as a brownish white amorphous solid after solvent removal. [M + 1] 412.1.

##### 4.1.2.4. 6-(4′-(3-Fluoro-4-methoxyphenyl)-[2,2′-bithiazol]-4-yl)-[1,3]dioxolo[4,5-b]pyridine, **d_4_**

**d_4_** was synthesized according to the general procedure (Section 4.1.2). First, 23.6 mg of 1-([1,3]dioxolo[4,5-b]pyridin-6-yl)-2-bromoethan-1-one dissolved in 500 µL of DMF acidified with 25 μL of glacial acetic acid was added to 20.0 mg (0.5948 mmol) of **b_4_** dissolved in 500 µL of DMF acidified with 25 μL of glacial acetic acid in a 10 mL glass vial. The mixture was stirred overnight at 45 °C. The resulting mixture was then transferred to a 10 mL falcon tube. The rest followed as described in Section 4.1.2.2. **d_4_** was obtained with ≥99% HPLC purity as a pale yellow amorphous solid after solvent removal. 

##### 4.1.2.5. 4-(Benzo[d][1,3]dioxol-5-yl)-4′-(6-fluoropyridin-3-yl)-2,2′-bithiazole, **d_6_**

**d_6_** was synthesized according to the general procedure (Section 4.1.2). First, 37.5 mg of bromo-1-(6-fluoropyridin-3-yl)ethan-1-one dissolved in 500 µL of DMF was added to 35.0 mg (0.1324 mmol) of **b_1_** dissolved in 500 µL of DMF solution in a 10 mL vial. The mixture was stirred overnight at 35 °C. The resulting mixture was transferred into a 10 mL falcon tube and centrifuged at 6000 rpm for 10 min. The resulting sediment was sonicated twice in 5 mL of MeOH for 10 min, each time followed by centrifugation for 10 min (6000 rpm) till 99% HPLC purity of **d_6_**. **d_6_** was obtained as an off-white amorphous solid after solvent removal. [M + 1] 384.0. 

##### 4.1.2.6. 6-(4′-(6-Fluoropyridin-3-yl)-[2,2′-bithiazol]-4-yl)-[1,3]dioxolo[4,5-b]pyridine, **d_8_**

**d_8_** was synthesized according to the general procedure (Section 4.1.2). First, 46.4 mg of 1-([1,3]dioxolo[4,5-b]pyridin-6-yl)-2-bromoethan-1-one dissolved in 500 µL of DMF acidified with 10 μL of glacial acetic acid was added to 35.0 mg (0.1463 mmol) of **b_7_** dissolved in 500 µL of DMF acidified with 15 μL of glacial acetic acid in a 10 mL glass vial. The mixture was stirred overnight at rt. The resulting mixture was transferred into a 10 mL falcon tube and centrifuged at 6000 rpm for 10 min. The resulting sediment was sonicated twice in 10 mL of a 1:1 mixture of acetone and methanol for 5 min, each time followed by centrifugation for 10 min (6000 rpm) till 99% HPLC purity of **d_8_**. **d_8_** was obtained as an off-white amorphous solid after solvent removal. [M + 1] 385.3. 

##### 4.1.2.7. 4-(4′-(2-Fluoropyrimidin-5-yl)-[2,2′-bithiazol]-4-yl)-[1,3]dioxolo[4,5-b]pyridine, **d_10_**

**d_10_** was synthesized according to the general procedure (Section 4.1.2). First, 38.0 mg of 1-([1,3]dioxolo[4,5-b]pyridin-6-yl)-2-bromoethan-1-one dissolved in 200 µL of DMF acidified with 10 μL of glacial acetic acid was added to 28.8 mg (0.1199 mmol) of **b_10_** dissolved in 400 µL of DMF acidified with 20 μL of glacial acetic acid in a 10 mL glass vial. The rest followed as described in Section 4.1.2.1. **d_10_** was obtained with ≥99% HPLC purity as a yellowish-white amorphous solid after solvent removal. [M + 1] 386.3. 

##### 4.1.2.8. 2-(4′-([1,3]Dioxolo[4,5-b]pyridin-6-yl)-[2,2′-bithiazol]-4-yl)pyrimidin-5-ol, **d_11_**

**d_11_** was synthesized according to the general procedure (Section 4.1.2). First, 170.0 mg of 1-([1,3]dioxolo[4,5-b]pyridin-6-yl)-2-bromoethan-1-one dissolved in 500 µL of DMF acidified with 25 μL of glacial acetic acid was added to 128.0 mg (0.5372 mmol) of **b_11_** dissolved in 500 µL of DMF acidified with 25 μL in a 10 mL glass vial. The mixture was stirred overnight at 45 °C. The resulting mixture was transferred into a 10 mL falcon tube and centrifuged at 6000 rpm for 10 min. The resulting sediment was sonicated twice in 10 mL of MeOH for 5 min, each time followed by centrifugation for 10 min (6000 rpm) till 99% HPLC purity of **d_11_**. **d_11_** was obtained as a brown amorphous solid after solvent removal. [M + 1] 384.2. 

##### 4.1.2.9. 2-(4′-(Benzo[d][1,3]dioxol-5-yl)-[2,2′-bithiazol]-4-yl)pyrimidin-5-ol, **d_12_**

**d_12_** was synthesized according to the general procedure (Section 4.1.2). First, 160.0 mg of 2-bromo-1-(5-hydroxypyrimidin-2-yl)ethan-1-one dissolved in 500 µL of DMF was added to 150.0 mg (0.5675 mmol) of **b_1_** dissolved in 500 µL of DMF in a 10 mL glass vial. The mixture was stirred overnight at 40 °C. The rest followed as described in Section 4.1.2.2. ≥99% HPLC pure **d_12_** was obtained as a yellowish amorphous solid. [M + 1] 383.2. 

##### 4.1.2.10. 4-(4′-([1,3]Dioxolo[4,5-b]pyridin-6-yl)-[2,2′-bithiazol]-4-yl)-2-fluorophenol, **d_13_**

**d_13_** was synthesized according to the general procedure (Section 4.1.2). First, 125.0 mg of 1-([1,3]dioxolo[4,5-b]pyridin-6-yl)-2-bromoethan-1-one dissolved in 500 µL of DMF acidified with 25 μL of glacial acetic acid was added to 100.0 mg (0.3932 mmol) of **b_12_** dissolved in 500 µL of DMF acidified with 100 μL of glacial acetic acid in a 10 mL glass vial. The mixture was stirred overnight at rt. The rest followed as described in Section 4.1.2.2. ≥99% HPLC pure **d_13_** was obtained as an off-white solid. [M + 1] 400.3. 

##### 4.1.2.11. 4-(4′-(Benzo[d][1,3]dioxol-5-yl)-[2,2′-bithiazol]-4-yl)phenol, **d_14_**

**d_14_** was synthesized according to the general procedure (Section 4.1.2). First, 87.5 mg of 2-bromo-1-(4-hydroxyphenyl)ethan-1-one dissolved in 500 µL of DMF acidified with 50 μL of glacial acetic acid then added to 75.4 mg (0.2853 mmol) of **b_1_** dissolved in 500 µL of DMF in a 10 mL glass vial. The mixture was stirred overnight at 40 °C. The resulting mixture was transferred into a 10 mL falcon tube and centrifuged at 6000 rpm for 10 min, after which it was transferred to a 10 mL falcon tube, and Milli-Q^®^ water was added to the mixture until a 10 mL mixture was attained, which resulted in the precipitation of **d_14_**. The mixture was left in the −20 °C fridge for 20 min, then was centrifuged at 15 °C for 10 min. The rest followed as described in Section 4.1.2.2. ≥99% HPLC pure **d_14_** was obtained as a white solid. [M + 1] 381.2. 

##### 4.1.2.12. 4-(4′-([1,3]Dioxolo[4,5-b]pyridin-6-yl)-[2,2′-bithiazol]-4-yl)phenol, **d_15_**

**d_15_** was synthesized according to the general procedure (Section 4.1.2). First, 47.1 mg of 1-([1,3]dioxolo[4,5-b]pyridin-6-yl)-2-bromoethan-1-one dissolved in 500 µL of DMF acidified with 100 μL of glacial acetic acid was added to 30.4 mg (0.1286 mmol) of **b_13_** dissolved in 300 µL of DMF acidified with 100 μL of glacial acetic acid in a 10 mL glass vial. An additional quantity of 50 μL of glacial acetic acid was added to the mixture. The mixture was stirred overnight at rt. The resulting mixture was transferred into a 10 mL falcon tube and centrifuged at 6000 rpm for 10 min. The rest followed as described in Section 4.1.2.2. ≥99% HPLC pure **d_15_** was obtained as an off-white amorphous solid. [M + 1] 381.2. 

##### 4.1.2.13. 4-(4′-([1,3]Dioxolo[4,5-b]pyridin-6-yl)-[2,2′-bithiazol]-4-yl)benzene-1,3-diol, **d_22_**

**d_22_** was synthesized according to the general procedure (Section 4.1.2). First, 54.2 mg of 1-([1,3]dioxolo[4,5-b]pyridin-6-yl)-2-bromoethan-1-one dissolved in 500 µL of DMF acidified with 100 μL of glacial acetic acid was added to 40 mg (0.1585 mmol) of **b_16_** dissolved in 500 µL of DMF acidified with 100 μL of glacial acetic acid in a 10 mL glass vial. The reaction was left to run overnight at 45 °C. The resulting mixture was transferred into a 10 mL falcon tube and centrifuged at 6000 rpm for 10 min. The rest followed as described in Section 4.1.2.2. ≥99% HPLC pure **d_22_** was obtained as an orange-yellow amorphous solid. [M + 1] 398.2. 

#### 4.1.3. Synthesis of the DABTAs from the Phenol and Pyrimidinol DABTAs

In this step, the phenol- and pyrimidinol-bearing DABTAs obtained from Step II of the chemical synthesis (Figure 2) were either O-fluoroethylated or O-fluoro/iodoPEGylated. The details involved for each of the ligands are presented below.

##### 4.1.3.1. 6-(4′-(5-(2-Fluoroethoxy)pyrimidin-2-yl)-[2,2′-bithiazol]-4-yl)-[1,3]dioxolo[4,5-b]pyridine, **f_4_**

K_2_CO_3_ (7.2 mg, 0.0521 mmol, 2.0 mol equiv) was added to 1000 µL of DMF which contained a suspension of 10 mg (0.0261 mmol) of **d_11_** in a 10 mL glass vial. The mixture was heated while stirring with a magnetic stirrer for 10 min at 80 °C. Subsequently, 14.1 µL (0.04937 mmol, 2.1 mol equiv) of TBAH (1 M) was added, and the mixture was stirred at rt until complete dissolution of the suspension. Subsequently, 2.5 µL (0.0339 mmol, 1.3 mol equiv) of 1-bromo-2-fluoroethane was added to the mixture, and it was heated while stirring at 80 °C. The reaction was stopped after the complete consumption of **d_11_**. Next, 10 µL of 1N NaOH was added to the mixture, and it was then transferred to a 10 mL falcon tube. Then, 10 mL of Milli-Q^®^ water was added to the mixture, and it was left for 10 min at 4 °C. A quantity of 1 mL of MeOH was added to the mixture, and it was centrifuged at 6000 rpm for 10 min. Iterative washing of the sediments was carried out using 5 mL of MeOH, with 2 min sonication, followed by centrifugation for 10 min (6000 rpm), yielding 99% HPLC purity. MeOH was completely evaporated in vacuo, and **f_4_** was obtained as a white amorphous solid. [M + 1] 430.3. 

##### 4.1.3.2. 4-(Benzo[d][1,3]dioxol-5-yl)-4′-(5-(2-fluoroethoxy)pyrimidin-2-yl)-2,2′-bithiazole, **f_5_**

K_2_CO_3_ (10.8 mg, 0.0784 mmol, 1.5 mol equiv) was added to 1000 µL of DMF which contained a suspension of 20 mg (0.0523 mmol) of **d_12_** in a 10 mL glass vial. The mixture was heated while stirring with a magnetic stirrer for 10 min at 80 °C. Subsequently, 19.2 µL (0.03654 mmol, 1.4 mol equiv) of TBAH (1 M) was added to the mixture, followed by 13.4 μL (0.07844 mmol, 1.5 mol equiv) of 2-fluoroethyl 4-methylbenzenesulfonate. The mixture was heated while stirring at 80 °C. The reaction was stopped after the complete consumption of **d_12_**. Further workup follows as described in Section 4.1.3.1. **f_5_** was obtained as a white waxy amorphous solid. [M + 1] 429.3. 

##### 4.1.3.3. 4-(Benzo[d][1,3]dioxol-5-yl)-4′-(5-(2-(2-fluoroethoxy)ethoxy)pyrimidin-2-yl)-2,2′-bithiazole, **f_6_**

K_2_CO_3_ (10.8 mg, 0.0784 mmol, 1.5 mol equiv) was added to 1000 µL of DMF which contained a suspension of 20 mg (0.0523 mmol) of **d_12_** in a 10 mL glass vial. The mixture was heated while stirring with a magnetic stirrer for 10 min at 80 °C. Subsequently, 19.2 µL (0.03654 mmol, 1.4 mol equiv) of TBAH (1 M) was added to the mixture, followed by 7.8 µL (0.0362 mmol, 1.5 mol equiv) of 1-bromo-2-(2-fluoroethoxy)ethane. The mixture was heated while stirring at 80 °C. The reaction was stopped after the complete consumption of **d_12_**. Further workup follows as described in Section 4.1.3.1. **f_6_** was obtained as a white amorphous solid. [M + 1] 473.3.

##### 4.1.3.4. 4-(Benzo[d][1,3]dioxol-5-yl)-4′-(5-(2-(2-(2-fluoroethoxy)ethoxy)ethoxy)pyrimidin-2-yl)-2,2′-bithiazole, **f_7_**

K_2_CO_3_ (10.8 mg, 0.0784 mmol, 1.5 mol equiv) was added to 1000 µL of DMF which contained a suspension of 20 mg (0.0523 mmol) of **d_12_** in a 10 mL glass vial. The mixture was heated while stirring with a magnetic stirrer for 10 min at 80 °C. Subsequently, 19.2 µL (0.03654 mmol, 1.4 mol equiv) of TBAH (1 M) was added to the mixture, followed by 12.0 µL (0.0784 mmol, 2.0 mol equiv) of 1-bromo-2-(2-(2-fluoroethoxy)ethoxy)ethane. The mixture was heated while stirring at 80 °C. The reaction was stopped after the complete consumption of **d_12_**. Further workup follows as described in Section 4.1.3.1. **f_7_** was obtained as a yellowish-white amorphous solid. [M + 1] 517.3. 

##### 4.1.3.5. 6-(4′-(5-(2-(2-Fluoroethoxy)ethoxy)pyrimidin-2-yl)-[2,2′-bithiazol]-4-yl)-[1,3]dioxolo[4,5-b]pyridine, **f_8_**

K_2_CO_3_ (7.2 mg, 0.0521 mmol, 2.0 mol equiv) was added to 1000 µL of DMF which contained a suspension of 10 mg (0.0261 mmol) of **d_11_** in a 10 mL glass vial. The mixture was heated while stirring with a magnetic stirrer for 10 min at 80 °C. Subsequently, 14.1 µL (0.04937 mmol, 2.1 mol equiv) of TBAH (1 M) was added, and the mixture was stirred at rt until complete dissolution of the suspension. Subsequently, 6.0 µL (0.05218 mmol, 2 mol equiv) of 1-bromo-2-(2-fluoroethoxy)ethane was added to the mixture, and it was heated while stirring at 80 °C. The reaction was stopped after the complete consumption of **d_11_**. Further workup follows as described in Section 4.1.3.1. **f_8_** was obtained as an off-white amorphous solid. [M + 1] 474.3. 

##### 4.1.3.6. 6-(4′-(3-(2-(2-Fluoroethoxy)ethoxy)-4-methoxyphenyl)-[2,2′-bithiazol]-4-yl)-[1,3]dioxolo [4,5-b]pyridine, **f_9_**

K_2_CO_3_ (10.1 mg, 0.0729 mmol, 2 mol equiv) was added to 1000 µL of DMF which contained a suspension of 15 mg (0.0364 mmol) of **d_3_** in a 10 mL glass vial. The mixture was heated while stirring with a magnetic stirrer for 5 min at 80 °C. Subsequently, 22.9 µL (0.1017 mmol, 3 mol equiv) of TBAH (1 M) was added to the mixture, followed by 10 μL (0.1001 mmol, 2 mol equiv) of 1-bromo-2-(2-fluoroethoxy)ethane. The mixture was heated while stirring at 80 °C. The reaction was stopped after the complete consumption of **d_3_**. Further workup follows as described in Section 4.1.3.1. **f_9_** was obtained as an off-white amorphous solid. 

##### 4.1.3.7. 6-(4′-(3-Fluoro-4-(2-(2-(2-fluoroethoxy)ethoxy)ethoxy)phenyl)-[2,2′-bithiazol]-4-yl)-[1,3]dioxolo[4,5-b]pyridine, **f_10_**

K_2_CO_3_ (10.4 mg, 0.07525 mmol, 1.5 mol equiv) was added to 800 µL of DMF which contained a suspension of 20 mg (0.0500 mmol) of **d_13_** in a 10 mL glass vial. The mixture was heated while stirring with a magnetic stirrer for 5 min at 80 °C. Subsequently, 14.2 µL (0.05010 mmol, 2 mol equiv) of TBAH (1 M) was added to the mixture, followed by 7.8 µL (0.0362 mmol, 1.5 mol equiv) of 1-bromo-2-(2-fluoroethoxy)ethane dissolved in 200 µL of DMF. The mixture was then heated while stirring at 80 °C. The reaction was stopped after the complete consumption of **d_13_**. Further workup follows as described in Section 4.1.3.1. **f_10_** was obtained as an off-white amorphous solid. [M + 1] 473.3. 

##### 4.1.3.8. 4-(Benzo[d][1,3]dioxol-5-yl)-4′-(4-(2-fluoroethoxy)phenyl)-2,2′-bithiazole, **f_11_**

K_2_CO_3_ (10.9 mg, 0.07887 mmol, 1.5 mol equiv) was added to 800 µL of DMF which contained a suspension of 20 mg (0.05257 mmol) of **d_14_** in a 10 mL glass vial. The mixture was heated while stirring with a magnetic stirrer for 5 min at 80 °C. Subsequently, 19.3 µL (0.07360 mmol, 1.5 mol equiv) of TBAH (1 M) was added to the mixture, followed by 16.7 μL (0.0592 mmol, 1.5 mol equiv) of 2-fluoroethyl 4-methylbenzenesulfonate. The mixture was heated while stirring at 80 °C. The reaction was stopped after the complete consumption of **d_14_**. Further workup follows as described in Section 4.1.3.1. **f_11_** was obtained as an off-white solid after lyophilization. [M + 1] 427.2. 

##### 4.1.3.9. 6-(4′-(4-(2-(2-(2-Fluoroethoxy)ethoxy)ethoxy)phenyl)-[2,2′-bithiazol]-4-yl)-[1,3]dioxolo[4,5-b]pyridine, **f_12_**

K_2_CO_3_ (5.4 mg, 0.03907 mmol, 1.5 mol equiv) was added to 400 µL of DMF which contained a suspension of 10 mg (0.02622 mmol) of **d_15_** in a 10 mL glass vial. The mixture was heated while stirring with a magnetic stirrer for 5 min at 80 °C. Subsequently, 13.7 µL (0.0524 mmol, 2 mol equiv) of TBAH (1 M) was added to the mixture, followed by 7.99 μL (0.0526 mmol, 2 mol equiv) of 1-bromo-2-(2-(2-fluoroethoxy)ethoxy)ethane. The mixture was heated while stirring at 80 °C. The reaction was stopped after the complete consumption of **d_15_**. Further workup follows as described in Section 4.1.3.1. **f_12_** was obtained as an off-white solid after lyophilization. [M + 1] 515.2. 

##### 4.1.3.10. 6-(4′-(5-(2-(2-(2-Fluoroethoxy)ethoxy)ethoxy)pyrimidin-2-yl)-[2,2′-bithiazol]-4-yl)-[1,3]dioxolo[4,5-b]pyridine, **f_13_**

K_2_CO_3_ (5.41 mg, 0.02608 mmol, 1.5 mol equiv) was added to 1000 µL of DMF which contained a suspension of 10 mg (0.02608 mmol) of **d_11_** in a 10 mL glass vial. The mixture was heated while stirring with a magnetic stirrer for 10 min at 80 °C. Subsequently, 14.1 µL (0.04937 mmol, 2.1 mol equiv) of TBAH (1 M) was added, and the mixture was stirred at rt until complete dissolution of the suspension. Next, 7.95 µL (0.0522 mmol, 2.0 mol equiv) of 1-bromo-2-(2-(2-fluoroethoxy)ethoxy)ethane was added to the mixture, and it was heated while stirring at 80 °C. The reaction was stopped after the complete consumption of **d_11_**. Then, 10 mL of Milli-Q^®^ water was added to the mixture, and it was left for 10 min at 4 °C, and 1 mL of MeOH was added to the mixture, and it was centrifuged at 6000 rpm for 10 min. Iterative washing of the precipitate was carried out using 5 mL of MeOH and acetone (5:1) with 1.7% glacial acetic acid added to the resulting sediment with 2 min sonication, followed by centrifugation for 10 min (6000 rpm) at 10 °C, yielding 99% HPLC purity. MeOH was completely evaporated in vacuo, and **f_13_** was obtained as an off-white solid after lyophilization. [M + 1] 518.2.

##### 4.1.3.11. 4-(Benzo[d][1,3]dioxol-5-yl)-4′-(3-(2-(2-(2-fluoroethoxy)ethoxy)ethoxy)-4-methoxyphenyl)-2,2′-bithiazole, **f_14_**

K_2_CO_3_ (6.74 mg, 0.0487 mmol, 2.0 mol equiv) was added to 500 µL of DMF which contained a suspension of 10 mg (0.02436 mmol) of **d_1_** in a 10 mL glass vial. The mixture was heated while stirring with a magnetic stirrer for 10 min at 80 °C. Subsequently, 12.5 µL (0.04871 mmol, 2.0 mol equiv) of TBAH (1M) was added to the mixture, followed by 7.42 µL (0.04873 mmol, 2.0 mol equiv) of 1-bromo-2-(2-(2-fluoroethoxy)ethoxy)ethane. The mixture was heated while stirring at 80 °C. The reaction was stopped after the complete consumption of **d_1_**. Further workup follows as described in Section 4.1.3.1. **f_14_** was obtained as an off-white solid after lyophilization. [M + 1] 545.2. 

##### 4.1.3.12. 4-(Benzo[d][1,3]dioxol-5-yl)-4′-(5-methoxypyrimidin-2-yl)-2,2′-bithiazole, **f_15_**

K_2_CO_3_ (8.13 mg, 0.0588 mmol, 1.5 mol equiv) was added to 500 µL of DMF which contained a suspension of 15 mg (0.0262 mmol) of **d_12_** in a 10 mL glass vial. The mixture was heated while stirring with a magnetic stirrer for 10 min at 80 °C. Subsequently, 9.6 µL (0.03654 mmol, 1.4 mol equiv) of TBAH (1M) was added to the mixture, followed by 7.1 µL (0.04704 mmol, 1.2 mol equiv) of methyltosylate. The mixture was heated while stirring at 80 °C. The reaction was stopped after the complete consumption of **d_12_**. Further workup follows as described in Section 4.1.3.1. **f_15_** was obtained as an off-white solid after lyophilization. [M + 1] 397.2. 

##### 4.1.3.13. 5-(4′-(5-Methoxypyrimidin-2-yl)-[2,2′-bithiazol]-4-yl)-[1,3]dioxolo[4,5-b]pyridine, **f_16_**

K_2_CO_3_ (5.41 mg, 0.02608 mmol, 1.5 mol equiv) was added to 1000 µL of DMF which contained a suspension of 10 mg (0.02608 mmol) of **d_11_** in a 10 mL glass vial. The mixture was heated while stirring with a magnetic stirrer for 10 min at 80 °C. Subsequently, 14.1 µL (0.04937 mmol, 2.1 mol equiv) of TBAH (1 M) was added, and the mixture was stirred at rt until complete dissolution of the suspension. Next, 5.1 µL (0.0339 mmol, 1.3 mol equiv) of methyltosylate was added to the mixture, and it was heated while stirring at 80 °C. The reaction was stopped after the complete consumption of **d_11_**. Further workup follows as described in Section 4.1.3.1. **f_16_** was obtained as a brown waxy amorphous solid after lyophilization. [M + 1] 398.3.

##### 4.1.3.14. 6-(4′-(4-Methoxyphenyl)-[2,2′-bithiazol]-4-yl)-[1,3]dioxolo[4,5-b]pyridine, **f_17_**

K_2_CO_3_ (5.4 mg, 0.03933 mmol, 1.5 mol equiv) was added to 500 µL of DMF which contained a suspension of 10 mg (0.02622 mmol) of **d_15_** in a 10 mL glass vial. The mixture was heated while stirring with a magnetic stirrer for 5 min at 80 °C. Subsequently, 13.7 µL (0.0524 mmol, 2 mol equiv) of TBAH (1 M) was added to the mixture, followed by 4.9 μL (0.07864 mmol, 2 mol equiv) of methyliodide. The mixture was heated while stirring at 80 °C. The reaction was stopped after the complete consumption of **d_15_**. Further workup follows as described in Section 4.1.3.1. **f_17_** was obtained as an off-white solid after lyophilization. [M + 1] 396.3.

##### 4.1.3.15. 6-(4′-(3,4-Dimethoxyphenyl)-[2,2′-bithiazol]-4-yl)-[1,3]dioxolo[4,5-b]pyridine, **f_18_**

K_2_CO_3_ (5.1 mg, 0.03645 mmol, 1.5 mol equiv) was added to 1000 µL of DMF which contained a suspension of 10 mg (0.02430 mmol) of **d_3_** in a 10 mL glass vial. The mixture was heated while stirring with a magnetic stirrer for 5 min at 80 °C. Subsequently, 12.7 µL (0.04860 mmol, 2 mol equiv) of TBAH (1 M) was added to the mixture, followed by 4.5 μL (0.07290 mmol, 2 mol equiv) of methyliodide. The mixture was heated while stirring at 80 °C. The reaction was stopped at 1 h when no further decrease in **d_3_** was observed. The reaction mixture (suspension) was transferred to a 10 mL falcon tube and centrifuged. Iterative washing of the resulting sediment was carried out first with 2 mL of DMF with 5 μL of TBAH (1 M), then with 2 mL of only DMF, and finally with 5 mL of acetone. The sediment was sonicated with the above solvents for 5 min, followed by centrifugation for 10 min (6000 rpm), yielding 99% HPLC purity. Acetone was completely evaporated in vacuo, and **f_18_** was obtained as an off-white amorphous solid. [M + 1] 426.2. 

##### 4.1.3.16. 6-(4′-(2,4-Dimethoxyphenyl)-[2,2′-bithiazol]-4-yl)-[1,3]dioxolo[4,5-b]pyridine, **f_19_**

K_2_CO_3_ (9.9 mg, 0.07171 mmol, 3.0 mol equiv) was added to 500 µL of DMF solution of **d_22_** (9.5 mg, 0.0239 mmol) in a 10 mL glass vial. The mixture was heated while stirring with a magnetic stirrer for 5 min at 80 °C. Subsequently, 18.8 µL (0.07168 mmol, 3.0 mol equiv) of TBAH (1 M) was added, followed by the addition of 6.0 µL (0.0956 mmol, 4 mol equiv) of methyliodide to the mixture, and it was heated while stirring at 80 °C. The reaction was stopped after the complete consumption of **d_22_**. Further workup follows as described in Section 4.1.3.1. **f_22_** was obtained as a dark brown amorphous solid. [M + 1] 426.2. 

##### 4.1.3.17. 6-(4′-(2,4-Bis(2-(2-fluoroethoxy)ethoxy)phenyl)-[2,2′-bithiazol]-4-yl)-[1,3]dioxolo[4,5-b]pyridine, **f_20_**

K_2_CO_3_ (7.0 mg, 0.05065 mmol, 2.0 mol equiv) was added to 1000 µL of DMF which contained a suspension of 10 mg (0.0216 mmol) of **d_22_** in a 10 mL glass vial. The mixture was heated while stirring with a magnetic stirrer for 10 min at 80 °C. Subsequently, 13.8 µL (0.05283 mmol, 2.1 mol equiv) of TBAH (1 M) was added, and the mixture was stirred at rt until complete dissolution of the suspension, followed by the addition of 12.2 µL (0.1006 mmol, 4 mol equiv) of 1-bromo-2-(2-fluoroethoxy)ethane to the mixture, and it was heated while stirring at 80 °C. The reaction was stopped after the complete consumption of **d_22_**. Further workup follows as described in Section 4.1.3.1. **f_20_** was obtained as a dark brown amorphous solid. [M + 1] 426.2.

##### 4.1.3.18. 6-(4′-(5-(2-(2-(2-Iodoethoxy)ethoxy)ethoxy)pyrimidin-2-yl)-[2,2′-bithiazol]-4-yl)-[1,3]dioxolo[4,5-b]pyridine, **h_13_**

K_2_CO_3_ (5.41 mg, 0.02608 mmol, 1.5 mol equiv) was added to 1000 µL of DMF which contained a suspension of 10 mg (0.02608 mmol) of **d_11_** in a 10 mL glass vial. The mixture was heated while stirring with a magnetic stirrer for 10 min at 80 °C. Subsequently, 14.1 µL (0.04937 mmol, 2.1 mol equiv) of TBAH (1 M) was added, and the mixture was stirred at rt until complete dissolution of the suspension. Next, 9.5 µL (0.0522 mmol, 2.0 mol equiv) of 1,2-bis(2-iodoethoxy)ethane was added to the mixture, and it was heated while stirring at 80 °C. The reaction was stopped after the complete consumption of **d_11_**. Further workup follows as described in Section 4.1.3.1. **h_13_** was obtained as an off-white solid after lyophilization. [M + 1] 426.2.

### 4.2. In Vitro Binding Assays

All chemicals were obtained from Merck KGaA (Darmstadt, Germany), unless stated otherwise.

#### 4.2.1. Preparation of the Amyloid Protein Aggregates

##### 4.2.1.1. Preparation of Recombinant α-syn Fibrils

The incubation of purified recombinant α-synuclein monomer (10 mg/mL) obtained as described by Uzuegbunam [1,22] was carried out in PBS (pH 7.4) at 37 °C. The solution was left shaking at 900 rpm in an Eppendorf Thermomixer for up to 84 h. Determination of the concentration of the formed fibrils (α-syn) was carried out following the centrifugation of the reaction vial at 12,000× *g* for 5 min. The pellet obtained was washed thrice with 20 mM of Tris-HCl, pH 7.4. The supernatants from the three centrifugation steps were combined, and the concentration of the monomer in the resulting supernatant was determined in a BCA assay (Thermo Fisher Scientific, Waltham, MA, USA) (with a BSA standard curve). Determination of the concentration of α-syn was carried out by subtracting the amount of α-synuclein monomer in the supernatant from the initial amount used for fibrillation.

##### 4.2.1.2. Preparation of Recombinant β-Amyloid Fibrils

After the dissolution of 1 mg of β-amyloid_1–42_ peptide (Abcam, Cambridge, UK) in 50 μL of DMSO, 925 μL of Milli-Q^®^ water was added to it, and the mixture was stirred. Finally, 25 μL of 1M Tris HCl, pH 7.4 was added to bring the final peptide concentration to 1 mg/mL. The mixture was then left to incubate at 37 °C for 30 h with shaking at 900 rpm. To separate the non-aggregated monomers from the formed fibrils, the reaction mixture was centrifuged at 12,000× *g* for 5 min, and the resulting pellet was washed three times with 500 μL of 20 mM Tris-HCl, pH 7.4. The supernatants from the three centrifugation steps were combined, and the concentration of the monomer in the resulting supernatant was determined in a BCA assay (Thermo Fisher Scientific, Waltham, MA, USA) (with a BSA standard curve). Determination of the concentration of fibrils formed was carried out by subtracting the amount of β-amyloid_1−42_ peptide monomer in the supernatant from the initial amount used for fibrillation.

##### 4.2.1.3. Preparation of Recombinant Tau Fibrils

Recombinant tau monomer (R&D systems, Minneapolis, MN, USA) was dissolved in 20 mM of Tris HCl, pH 8.0, 100 mM of NaCl, 25 μM of low-MW heparin, and 0.5 mM of dithiothreitol to a final concentration of 300 µg/mL. The mixture was incubated with shaking at 900 rpm at 37 °C for 48 h. To separate the non-aggregated monomers from the formed fibrils, the reaction mixture was centrifuged at 12,000× *g* for 5 min, and the resulting pellet was washed three times with 500 μL of 20 mM Tris-HCl, pH 7.4. The supernatants from the three centrifugation steps were combined, and the concentration of the monomer in the resulting supernatant was determined in a BCA assay(Thermo Fisher Scientific, Waltham, MA, USA) (with a BSA standard curve). Determination of the concentration of fibrils formed was carried out by subtracting the amount of tau peptide monomer in the supernatant from the initial amount used for fibrillation.

#### 4.2.2. Preparation of α-Synuclein, β-Amyloid_1−42_, and Tau Fibrils for Competition Binding Assays

The pellets of the fibrils obtained above (Section 4.2.1) were resuspended in 20 mM of Tris-HCl, pH 7.4 and diluted as needed for the binding assays.

#### 4.2.3. Competition Binding Assays

The binding assays were carried out with both tritiated DCVJ ([^3^H]DCVJ) (Novandi Chemistry AB, Södertälje, Sweden) and tritiated PiB ([^3^H]PiB) (Figure 6). [^3^H]DCVJ binds to all the protein aggregates with high affinity: K_d_ for α-syn 4.4 nM, Aβ 8.9 nM, tau 15.6 nM. [^3^H]PiB also binds with relatively high affinity to the fibrils [23,24,25]: K_d_ for α-syn 4.9 nM, Aβ 3.7 nM, tau 117.5 nM.

##### 4.2.3.1. Competition Binding Assays with the α-syn Fibrils

A fixed concentration of [^3^H]DCVJ (10 nM) and α-synuclein aggregates (125 nM), prepared as described above (Section 4.2.3), was used with different final concentrations of the DABTAs from 0.01 to 100 nM (see Appendix A). The experiment was carried out in quintuplicate for each concentration. The mixture was incubated for 2 h in a final volume of 200 μL per well after dilution with 20 mM of Tris-HCl, pH 7.4. The incubated mixtures were filtered through a Perkin Elmer GF/B glass filter via a TOMTEC cell harvester and immediately washed thrice with 1 mL of deionized water. The filters containing the bound ligands were dried and afterwards mixed with 3 mL of BetaPlate Scint solution (Perkin Elmer Life Sciences, Waltham, MA, USA) and left there for 2 h before counting in a Wallac 1450 MicroBeta TriLux Liquid Scintillation Counter (PerkinElmer Life Sciences, Waltham, MA, USA). For the determination of the inhibition constants, data were analyzed using GraphPad Prism software (version 7.0) to obtain IC_50_ values by fitting the data to the equation Y = bottom + (top − bottom)/(1 + 10(x − log IC_50_)). K_i_ values were calculated from IC_50_ values using the equation K_i_ = IC_50_/(1 + [radioligand]/K_d_). For assays with [^3^H]PiB, a fixed concentration of [^3^H]PiB (10 nM) and a 10 mM phosphate buffer (pH 7.4), 1 mM EDTA, and 0.5% BSA (for incubation and washing) were used instead. Further protocol was carried out as for [^3^H]DCVJ (see Appendix A).

##### 4.2.3.2. Competition Binding Assays for Aβ_1−42_ and Tau Fibrils

Fixed concentrations (125 nM) of Aβ_1−42_, tau aggregates, and [^3^H]DCVJ (10 nM and 20 nM for the Aβ_1−42_ and tau competition experiments respectively) were used with different concentrations of cold versions of the tracers from 1 to 1000 nM (See Appendix A). The experiment was performed in quadruplicate for each concentration. Further procedures followed the same already described in Section 4.2.3.1.

For assays with [^3^H]PiB, fixed concentrations (125 nM) of Aβ_1−42_, tau aggregates, and [^3^H]PiB (10 nM and 200 nM for the Aβ_1−42_ and tau competition experiments, respectively) with 10 mM phosphate buffer (pH 7.4) that contained 1 mM EDTA and 0.5% BSA (for incubation and washing) were used instead. Further procedures followed the same already described in Section 4.2.3.1 (See Appendix A).

## 5. Conclusions

The reported DABTAs based on in silico design showed impressive binding affinity to recombinant α-syn fibrils and selectivity over recombinant Aβ and tau aggregations in vitro. The binding details of these ligands to α-syn have been evaluated in detail with structure–activity relationships established in this paper. The introduction of 2–3 repeated PEGylation on the para-position of the benzene/pyrimidine ring was found to increase the binding affinities to α-syn as they facilitate increased π–π interactions. The lipophilicity of the newly reported ligands holds the promise of good brain pharmacokinetics. In order to further evaluate the efficacy of these ligands, they will be radiolabeled with either fluorine-18 or carbon-11 for further in vitro and in vivo studies.

## Data Availability

Data are contained within the article and Appendix A.

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
