# Peer review of "In Silico and In Vitro Study towards the Rational Design of 4,4′-Disarylbisthiazoles as a Selective α-Synucleinopathy Biomarker"

_ijms, 2023, doi:10.3390/ijms242216445_

Round 1

Reviewer 1 Report

Comments and Suggestions for Authors

This article discovered the potential of DABTA as an α-syn tracer through in vitro binding assays and computer simulations. This discovery may serve as a valuable tool to advance our comprehension of disease progression and, potentially, to monitor therapeutic effectiveness in clinical trials. The manuscript is of high quality.

Author Response

The authors would like to thank the reviewer for the positive evaluation.

Yours sincerely,

Behrooz H. Yousefi

Reviewer 2 Report

Comments and Suggestions for Authors

Author Response

The authors would like to thank the reviewer for the suggested changes and the positive evaluation. 

We have made all the following important changes in the revised version. 
â–ª The synthetic scheme is clearly presented in the main manuscript so that 
readers can easily understand the synthetic approach in the scheme. 
â–ª The structures of all compounds have been shown in a ChemDraw illustration 
instead of in the column of the table. 
â–ª The synthetic procedure and analytical details have been included in the main manuscript. 
Manuscript included. 
â–ª Only the 1H and 13C NMR spectra of the compounds were included in the 
Supporting Information.

Finally, we have added the selectivity results to Table 2 since the structures were removed.

We hope that we have been able to address all the key concerns of the esteemed reviewer.

Yours sincerely,

Behrooz H. Yousefi

Round 2

Reviewer 2 Report

Comments and Suggestions for Authors

The manuscript has been sufficiently improved.